# Prevalence and determinants of inadequately controlled diabetes in Qatar's public primary care settings: A cross-sectional study

Ahmed Sameer Alnuaimi *, Muslim Abbas Syed, Mohamed Ahmed Syed

Department of Clinical Research, Clinical Affairs Directorate, Primary Health Care Corporation, Qatar

* asalnuaimi@phcc.gov.qa

## Abstract

### Background

The prevalence of type 2 diabetes mellitus (T2DM) is rapidly increasing worldwide, causing serious morbidity and mortality. Proper management and control of T2DM are essential in slowing and reducing complications. Measuring the adequacy of diabetic control, therefore is one of the ways of evaluating the success of health care system and the individuals in addressing disease morbidity and mortality. This study was proposed to assess the extent of uncontrolled T2DM in ambulatory adults managed in the public primary care settings of Qatar over five years. In addition, it aimed to predict the risk of uncontrolled disease based on age, sex, nationality and comorbidity.

### Methods

A cross sectional review of Electronic Health Records of 84,512 adult individuals with T2DM having at least one HbA1c blood test result per year during the 5 years study period (1st January 2018, to 31st December 2022) was analyzed. The term "inadequately controlled" T2DM was used to define an individual with HbA1c measurements of 7% or higher.

### Results

The yearly prevalence of inadequately controlled T2DM remained stable at a high rate (53%) during the last three years of observations. Aging increases the probability of having inadequately controlled T2DM, but this was attributed to comorbidities. Adjusting for those comorbidities will show that younger ages are the ones at higher risk of being uncontrolled. Additionally, the study reported that hypertension and dyslipidemia are the most frequently prevalent comorbidities, nevertheless retinopathy, although less frequent, was the most important predictor for inadequately controlled T2DM.

**Data availability statement:** In compliance with the data privacy act in Qatar, the authors cannot legally distribute the anonymized source data used in this manuscript. However, interested researchers may submit a data request to the Department of Clinical Research in the Primary Health Care Corporation (PHCC) Qatar using the email researchsection@phcc.gov.qa. The data set used in the current manuscript was approved by the PHCC's Institutional Review Board (IRB reference number BUHOOTH-D-23-00055 on 13/08/2023).

**Funding:** The Primary Health Care Corporation will cover the publication fees incurred by the journal. The funder had no role in study design, data collection and analysis, decision to publish, or preparation of the manuscript.

**Competing interests:** The authors have declared that no competing interests exist.

## Conclusions

The study highlights the significant healthcare challenge posed by T2DM in Qatar due to its high prevalence and associated complications. Males, South Asian ethnic background and Qatari locals have the highest probability of being inadequately controlled. Overall, the study emphasizes the need for targeted interventions to address the challenges of controlling diabetes, particularly among high-risk groups identified in the study

## Introduction

Type 2 diabetes mellitus (T2DM) accounts for nearly 90% of the approximately 537 million diabetic cases worldwide [1]. Alarmingly, the prevalence of this disease is rapidly increasing among children and young adults [1–3]. Evidence indicates that this chronic condition is associated with serious health complications, negatively impacting the overall quality of life and mental health of those affected [4,5]. Despite recent advancements in diagnosis, treatment, and management, the hospitalization and mortality rates related to diabetes remain high [6,7]. Evidence suggests that early detection and proactive management are critical for preventing complications, particularly microvascular and macrovascular and reducing the associated mortality burden [7,8].

The risk factors affecting T2DM management and control are categorized into modifiable and non-modifiable [9]. Modifiable risk factors include diet, physical activity, alcohol and tobacco consumption, and compliance with diabetes medication [9]. Non-modifiable risk factors, as well as those related to social determinants of health, include age, gender, race, insurance status, marital status, comorbidities (such as retinopathy, neuropathy, nephropathy, coronary artery disease, cerebrovascular problems, hypertension, and dyslipidemia), duration of diabetes, educational level, source of diabetes education, family history of diabetes, and body mass index [10–13].

The management goal for all adults with diabetes is to maintain good glycemic control (optimal blood sugar level) in order to prevent macro and microvascular complications [12]. Measuring glycosylated hemoglobin (HbA1c) is the gold standard for assessing glycemic control. The term "uncontrolled diabetes" is defined as HbA1c levels higher than the set goal, being greater than 7% for over a six consecutive month period [14–16]. Evidence suggests that complications are more likely to occur in those with uncontrolled T2DM [6,17]. The Middle East and North Africa (MENA) region has the second-highest prevalence of diabetes worldwide [18]. In Qatar, T2DM is the most common non-communicable disease among Qataris (23%) and non-Qataris (18.3%) population.

Despite the availability of adequately equipped health care systems in the six GCC (Gulf Cooperation Council) countries, inadequately controlled DM continues to pose an important health challenge. Literature reported figures between 54.9% in Qatar [19] to as high as 77% in KSA [20] for the prevalence rate of inadequately controlled diabetes.

This study aimed to investigate the magnitude of uncontrolled T2DM in Qatar's the public primary care settings over a period of five years. The objective was to establish the association between risk factors and the severity of uncontrolled disease among adult PHCC service users with T2DM.

## Methods

Primary Health Care Corporation (PHCC) is Qatar's largest publicly funded primary care provider in Qatar. PHCC operates 31 health centers across the country, all of which are accredited by Accreditation Canada International [21]. All legal residents registered with PHCC are eligible to access its services. PHCC has an advanced electronic health record (EHR) system that was established in 2017. Data for the purposes of this study was extracted from PHCC's EHR.

### Study settings

The study population included all individuals aged 18 years and above diagnosed with T2DM, with at least one HbA1c test result per calendar year during the study period 1st January 2018–31st December 2022. Data points for the initial six months after T2DM diagnosis were excluded. This was done to provide ample time to establish appropriate management protocols following initial diagnosis.

### Study population

PHCC EHR uses SNOMED codes, a systematically organized collection of medical terms providing codes, terms, synonyms, and definitions used in clinical documentation and reporting to register all clinically relevant data [22]. These codes allowed the identification of seven comorbidities: dyslipidemia, hypertension, retinopathy, acute coronary disease, neuropathy, chronic kidney disease, and cardiovascular disease. Summing positive comorbidities per individual created a new variable (number of comorbidities) to reflect the health burden.

### Study variables

HbA1c is a form of hemoglobin (Hb) chemically linked to sugar. It measures a three-month average blood sugar level assessing glycemic control in individuals with diabetes [23]. According to the American Diabetes Association (ADA), good diabetic control for non-pregnant adults is defined as an HbA1c measurement below 7%, provided no significant hypoglycemia occurs [24].

As per ICD-10-CM, the term 'uncontrolled' T2DM can be classified into hypo and hyperglycemia [25]. For the purposes of this study the sub-classification of hyperglycemia was considered and the term "inadequately controlled" T2DM was used to define an individual with HbA1c measurements of 7% or higher based on previous studies [11,26,27].

As per PHCC's clinical guidelines on T2DM management, individuals may be subject to multiple HbA1C tests annually. For the purposes of the study the mean annual HbA1C value was considered. Only individuals with mean annual HbA1C values for more than one year (2–5 years) were included in the study.

To gain an understanding of an individual's overall T2DM control experience over the entire study period, a HbA1C diabetes control summary index was calculated to model diabetic individuals who were inadequately controlled for at least 50% of years in the follow up period. Therefore, individuals with only one year of follow up were excluded from calculating the aforementioned index.

Individual's nationalities were categorized into seven geographical regions as is described in the supplementary file [28]. To ensure valid statistical analysis, nationalities less than 300 individuals were grouped together as 'other'.

Data was extracted from PHCC's EHR for a five-year period (1st January 2018, to 31st December 2022). The study authors, in collaboration with the PHCC's department of Business Health Intelligence (BHI), were responsible for data collection and extraction. The data extraction was completed on 10/01/2024 using custom made filters by the BHI. These

data filters were thoroughly tested for performance. The data analysis and reporting followed the RECORD statement guidelines, which describe reporting standards for studies conducted using routinely collected health data [29].

### Data collection and quality assurance measures

The Statistical Package of Social Sciences (IBMSPSS Ver 28) was used for data analysis. Data cleaning involved logical checks for consistency of related variables and range checks for dates within the specified study period. Descriptive statistics summarized the demographic characteristics of the study population, including age, gender, and nationality. Proportions provided an overview of the participants' profiles. The association between the explanatory variables and the probability of inadequately controlled diabetes was tested at the bivariate level using the Prevalence Ratio (PR) for effect size and the Chi-square test for statistical significance. A multivariate model (Multiple Logistic Regression) was used to investigate the impact of different factors on the likelihood of being inadequately controlled, identifying significant predictors and their relative contributions. A P value <0.05 was considered statistically significant. The extremely large sample size allowed for an overpowered analysis, with a study power of 99% for an alpha of 0.05 and sample size of 60,735 (those with valid estimate of the outcome) to detect an adjusted Odds Ratio (multiple logistic regression) as small as 1.2 [30].

## Data analysis

### Ethical consideration

The study presented minimal risk of harm to its human participants. Patients registering with PHCC automatically consent to their records being used for service auditing, quality improvement, and research purposes. The data extracted from health records were anonymized ensuring that none of the participants' personal information was revealed to the research team. The study was conducted with integrity adhering to generally accepted ethical principles and was approved by the PHCC's Institutional Review Board (IRB reference number BUHOOTH-D-23–00055 on 13/08/2023).

### Patient and public involvement

There was no direct intervention/interaction with study participants. Patients' priorities, experience and preferences were not gathered nor were they involved in designing the study. There are no plans to disseminate results to the study participants directly as they were anonymized.

## Results

A total of 84,512 individuals diagnosed with T2DM with at least one HbA1c test result during the five-year study period met the study' eligibility criteria (Table 1). This studied population represents 81.2% of all active (at least one clinical visit a year) PHCC registered population with a verified diagnosis of T2DM. Approximately a quarter had one year of recorded HbA1c results. Another fifth had two years of follow-up, while only 15.8% of extracted records had data covering the full five-year study period.

**Table 1. HbA1c results by duration of follow up.**

| HbA1C follow up (years) | N | % |
| --- | --- | --- |
| 1 | 23777 | 28.1 |
| 2 | 18039 | 21.3 |
| 3 | 15297 | 18.1 |
| 4 | 14078 | 16.7 |
| 5 | 13321 | 15.8 |
| **Total** | **84512** | **100.0** |

Of the eligible population, those 70 + years and below 50 years old constituted 33.7% and 12% respectively. More than half (55.9%) were male. Southern Asia and Qatari nationalities comprised two thirds (67.4%) of the study population. South-eastern Asia, Sub-Saharan Africa, and 'Other' categories were the least represented accounting for only 6% of the studied population, Table 2.

Dyslipidemia and hypertension were the most frequent, each affecting 67.2% and 63% of individuals diagnosed with T2DM, Table 3. Retinopathy, acute coronary diseases and neuropathy ranged between 10.3–17.3% and ranked third.

A trend analysis of inadequately controlled diabetes (defined as an average HbA1C≥7%) by calendar year showed a reduction from 58.8% in 2018 to 53.2% in 2020 and plateaued over the following two years, Fig 1.

The probability of inadequately controlled T2DM among males was 24% higher compared to females (see Table 4). For the 50–69 years age group, the probability remained similar compared to the 70 + year age group. However, there

**Table 2. Sociodemographic characteristics of study population.**

|  | N | % |
|---|---|---|
| Age group (years) |  |  |
| 18-39 | 9,327 | 11.0 |
| 40-49 | 19,213 | 22.7 |
| 50-59 | 25,094 | 29.7 |
| 60-69 | 20,761 | 24.6 |
| 70+ | 10,117 | 12.0 |
| Total | 84,512 | 100.0 |
| Gender |  |  |
| Female | 37,304 | 44.1 |
| Male | 47,208 | 55.9 |
| Total | 84,512 | 100.0 |
| Nationality groups |  |  |
| Southern Asia | 29,545 | 35.0 |
| Qatari | 27,351 | 32.4 |
| Northern Africa | 12,016 | 14.2 |
| Western Asia | 10,516 | 12.4 |
| South-eastern Asia | 3213 | 3.8 |
| Sub-Saharan Africa | 991 | 1.2 |
| Others | 880 | 1.0 |
| Total | 84,512 | 100.0 |

**Table 3. Prevalence of selected comorbidities.**

| Comorbidities (Total N = 84512) | N | % |
|---|---|---|
| Dyslipidemia | 56801 | 67.2 |
| Hypertension | 53265 | 63 |
| Retinopathy | 14581 | 17.3 |
| Acute Coronary Disease | 9734 | 11.5 |
| Neuropathy | 8681 | 10.3 |
| Chronic kidney disease | 6996 | 8.3 |
| Cardiovascular disease | 2996 | 3.5 |

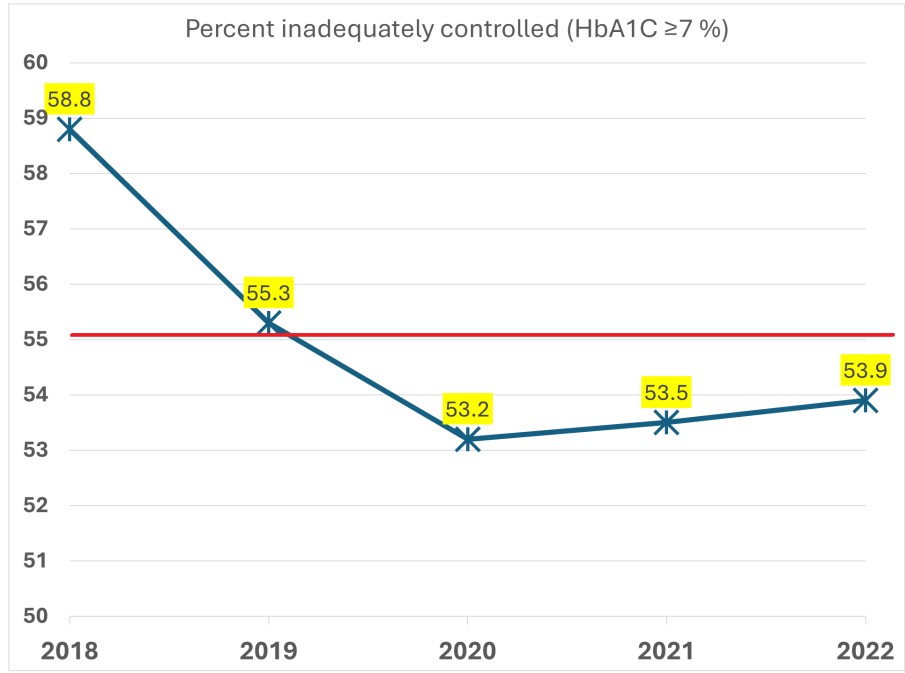

**Fig 1. Line graph showing the rate of inadequately controlled diabetes (HbA1c≥7%) by calendar year.**

was a statistically significant reduction in risk amongst 18–39 (19%) and 40–49-year (10%) age groups in comparison to 70 + year age group.

The prevalence of inadequately controlled T2DM was the lowest at 32.7% in the 'Other' nationality group. The probability of inadequately controlled T2DM amongst Southern Asia and Qatari nationality groups was the highest in comparison the 'Other' nationality group at 82% and 52% respectively. The risk estimate ranged from 29%−41% in other nationality groups.

Among the seven comorbidities considered in this study, retinopathy was associated with the highest probability (51%) of having inadequately controlled T2DM at the same time compared to its absence. Summing up all the comorbidities in a score demonstrated a clear positive trend in the probability of inadequately controlled T2DM with an increased score. Compared to those with no comorbidities (zero score) the probability increased by 19% for those with only one comorbidity and was two times higher for those with five to seven comorbidities (See Table 4).

A multiple logistic regression model was used to assess the net and independent association of four explanatory variables with the risk of inadequate T2DM control over the five-year study period. The model was statistically significant and accurately classified the group membership for the outcome with 61.6% accuracy. After adjusting for gender, nationality, and the comorbid conditions, the probability of inadequately controlled T2DM showed a statistically significant negative trend with age. The youngest age group had the highest increase in risk estimate at 48% compared to the oldest age group (70 + years). After adjusting for the possible confounding effects of the remaining independent variables included in the model, the male gender significantly increased the probability of inadequately controlled T2DM by 44% compared to females. The "Other" nationality group (European /North American /Miscellaneous) had the lowest rate of inadequately controlled T2DM and served as the reference category for assessing the risk estimate of the outcome. The Southern Asia nationality group was associated with the highest increase in risk estimate by 2.95 times, followed by Qatari and Sub-Saharan Africa nationality, each with around 2 times increase in risk estimate. The North Africa nationality group was

**Table 4. Risk of inadequately controlled T2DM (at least 50% of times throughout the study period) sociodemographic characteristics and comorbidities.**

| | | Inadequately controlled DM (at least 50% of years included) | | | | | | Prevalence Ratio (PR) | 95% CI PR | P |
|---|---|---|---|---|---|---|---|---|---|---|
| | | **Negative** | | **Positive** | | **Total** | | | | |
| | | **N** | **%** | **N** | **%** | **N** | **%** | | | |
| 1. | Gender | | | | | | | | | |
| | Female | 15157 | 55.1 | 12341 | 44.9 | 27498 | 100 | Ref | | |
| | Male | 14688 | 44.2 | 18549 | 55.8 | 33237 | 100 | 1.24 | (1.22 - 1.26) | <0.001 |
| 2. | Age group (years) | | | | | | | | | |
| | 70+ | 3794 | 47.2 | 4243 | 52.8 | 8037 | 100 | Ref | | |
| | 60-69 | 7539 | 46.3 | 8727 | 53.7 | 16266 | 100 | 1.02 | (0.99 - 1.05) | 0.66[NS] |
| | 50-59 | 8820 | 47.9 | 9585 | 52.1 | 18405 | 100 | 0.99 | (0.97 - 1.01) | 0.77[NS] |
| | 40-49 | 6644 | 52.4 | 6032 | 47.6 | 12676 | 100 | 0.9 | (0.88 - 0.93) | <0.001 |
| | 18-39 | 3048 | 57 | 2303 | 43 | 5351 | 100 | 0.81 | (0.78 - 0.84) | <0.001 |
| 3. | Nationality groups | | | | | | | | | |
| | Others | 381 | 67.3 | 185 | 32.7 | 566 | 100 | Ref | | |
| | Southern Asia | 8267 | 40.5 | 12126 | 59.5 | 20393 | 100 | 1.82 | (1.62 - 2.05) | <0.001 |
| | Qatari | 10461 | 50.2 | 10371 | 49.8 | 20832 | 100 | 1.52 | (1.35 - 1.71) | <0.001 |
| | Sub-Saharan Africa | 360 | 53.8 | 309 | 46.2 | 669 | 100 | 1.41 | (1.22 - 1.63) | <0.001 |
| | Western Asia | 4238 | 55.4 | 3412 | 44.6 | 7650 | 100 | 1.36 | (1.21 - 1.53) | <0.001 |
| | Northern Africa | 5010 | 57.8 | 3664 | 42.2 | 8674 | 100 | 1.29 | (1.14 - 1.46) | <0.001 |
| | South-eastern Asia | 1128 | 57.8 | 823 | 42.2 | 1951 | 100 | 1.29 | (1.13 - 1.47) | <0.001 |
| 4. | Hypertension | | | | | | | | | |
| | Negative | 11362 | 56.7 | 8675 | 43.3 | 20037 | 100 | Ref | | |
| | Positive | 18483 | 45.4 | 22215 | 54.6 | 40698 | 100 | 1.26 | (1.24 - 1.28) | <0.001 |
| 5. | Dyslipidemia | | | | | | | | | |
| | Negative | 9377 | 59.5 | 6373 | 40.5 | 15750 | 100 | Ref | | |
| | Positive | 20468 | 45.5 | 24517 | 54.5 | 44985 | 100 | 1.35 | (1.32 - 1.38) | <0.001 |
| 6. | Cardiovascular disease | | | | | | | | | |
| | Negative | 28937 | 49.4 | 29633 | 50.6 | 58570 | 100 | Ref | | |
| | Positive | 908 | 41.9 | 1257 | 58.1 | 2165 | 100 | 1.15 | (1.11 - 1.19) | <0.001 |
| 7. | Neuropathy | | | | | | | | | |
| | Negative | 26799 | 49.8 | 26978 | 50.2 | 53777 | 100 | Ref | | |
| | Positive | 3046 | 43.8 | 3912 | 56.2 | 6958 | 100 | 1.12 | (1.1 - 1.15) | <0.001 |
| 8. | Chronic kidney disease | | | | | | | | | |
| | Negative | 27737 | 50.1 | 27593 | 49.9 | 55330 | 100 | Ref | | |
| | Positive | 2108 | 39 | 3297 | 61 | 5405 | 100 | 1.22 | (1.19 - 1.25) | <0.001 |
| 9. | Retinopathy | | | | | | | | | |
| | Negative | 26141 | 53.8 | 22417 | 46.2 | 48558 | 100 | Ref | | |
| | Positive | 3704 | 30.4 | 8473 | 69.6 | 12177 | 100 | 1.51 | (1.49 - 1.53) | <0.001 |
| 10. | Acute Coronary Disease | | | | | | | | | |
| | Negative | 26994 | 50.3 | 26644 | 49.7 | 53638 | 100 | Ref | | |
| | Positive | 2851 | 40.2 | 4246 | 59.8 | 7097 | 100 | 1.2 | (1.18 - 1.23) | <0.001 |
| 11. | Number of comorbidities | | | | | | | | | |
| | 0 | 4695 | 63.6 | 2690 | 36.4 | 7385 | 100 | Ref | | |
| | 1 | 8264 | 56.8 | 6292 | 43.2 | 14556 | 100 | 1.19 | (1.16 - 1.22) | <0.001 |
| | 2 | 10179 | 48.9 | 10618 | 51.1 | 20797 | 100 | 1.4 | (1.37 - 1.43) | 0.07[NS] |

*(Continued)*

**Table 4.** (Continued)

| | | Inadequately controlled DM (at least 50% of years included) | | | | | | Prevalence Ratio (PR) | 95% CI PR | P |
|---|---|---|---|---|---|---|---|---|---|---|
| | | Negative | | Positive | | Total | | | | |
| | | N | % | N | % | N | % | | | |
| | 3 | 4567 | 40.5 | 6714 | 59.5 | 11281 | 100 | 1.63 | (1.59 - 1.67) | <0.001 |
| | 4 | 1583 | 34.2 | 3042 | 65.8 | 4625 | 100 | 1.81 | (1.76 - 1.86) | <0.001 |
| | 5 | 438 | 27.3 | 1167 | 72.7 | 1605 | 100 | 2 | (1.93 - 2.07) | <0.001 |
| | 6 | 110 | 25.3 | 325 | 74.7 | 435 | 100 | 2.05 | (1.93 - 2.17) | <0.001 |
| | 7 | 9 | 17.6 | 42 | 82.4 | 51 | 100 | 2.26 | (1.99 - 2.57) | <0.001 |

associated with the smallest increase in probability at 45% compared to the reference nationality category. Among the comorbidities, retinopathy was associated with the highest probability of being inadequately controlled, showing a statistically significant increase in risk estimate by more than two times (see Table 5). In contrast, chronic kidney disease and acute coronary disease were linked to a marginal increase in the calculated risk estimate of only 15%. Cardiovascular disease did not significantly contribute to the probability of having the outcome (See Table 5).

**Table 5.** A multiple logistic regression model with the risk of being inadequately controlled (at least 50% of times throughout the study period) as the dependent (response) variable by age, sex, nationality, and selected comorbidities as explanatory variables.

| | Adjusted OR | 95% Confidence Interval OR | P |
|---|---|---|---|
| 1. Age group (years) | | | <0.001 |
| The youngest age group (18–39 years of age) compared to the oldest age group (70+) | 1.48 | (1.37 to 1.61) | <0.001 |
| 40-49 years of age compared to the oldest age group (70+) | 1.35 | (1.27 to 1.44) | <0.001 |
| 50-59 years of age compared to the oldest age group (70+) | 1.38 | (1.31 to 1.47) | <0.001 |
| 60-69 years of age compared to the oldest age group (70+) | 1.25 | (1.18 to 1.32) | <0.001 |
| 2. Male gender compared to female | 1.44 | (1.39 to 1.49) | <0.001 |
| 3. Nationality groups | | | <0.001 |
| Southern Asia compared to Others (European/North America/ Miscellaneous) | 2.95 | (2.45 to 3.53) | <0.001 |
| Qatari nationality compared to Others (European/North America/ Miscellaneous) | 2.02 | (1.68 to 2.42) | <0.001 |
| Sub-Saharan Africa compared to Others (European/North America/ Miscellaneous) | 1.95 | (1.54 to 2.48) | <0.001 |
| South-eastern Asia compared to Others (European/North America/ Miscellaneous) | 1.73 | (1.41 to 2.12) | <0.001 |
| Western Asia compared to Others (European/North America/ Miscellaneous) | 1.61 | (1.34 to 1.94) | <0.001 |
| Northern Africa compared to Others (European/North America/ Miscellaneous) | 1.45 | (1.21 to 1.75) | <0.001 |
| 4. Comorbid condition | | | |
| Having Retinopathy | 2.59 | (2.47 to 2.71) | <0.001 |
| Having Dyslipidemia | 1.52 | (1.46 to 1.58) | <0.001 |
| Having Hypertension | 1.24 | (1.19 to 1.29) | <0.001 |
| Having Neuropathy | 1.17 | (1.11 to 1.24) | <0.001 |
| Having Chronic kidney disease | 1.15 | (1.08 to 1.23) | <0.001 |
| Having Acute Coronary Disease | 1.15 | (1.09 to 1.22) | <0.001 |
| Having Cardiovascular disease | 1.05 | (0.96 to 1.15) | 0.31[NS] |
| Constant | 0.16 | | <0.001 |

**Overall model classification accuracy=61.6%**

**P (Model) < 0.001**.

## Discussion

T2DM presents a significant healthcare challenge in Qatar due to its high prevalence and associated complications. This is the first study that highlights the prevalence of inadequately controlled T2DM and its determinants amongst adults in a Qatar's primary health care setting.

The study found that the yearly prevalence of inadequately controlled T2DM remained almost stable at around 53% during the last three years of observations, following an initial reduction in the first two years. Other studies from Qatar estimated a much higher rate of 86% [6,31], which may simply point out to different approaches for defining the outcome. The present study used the yearly average of HbA1c test results to define the outcome in the least biased way. Changing the selected HbA1c to the minimum, maximum or the last available value in a specific year would introduce significant fluctuations in the calculated prevalence rates. Additionally, evidence suggests that the fluctuating prevalence of uncontrolled T2DM over the course of time due to factors such as nature of disease progression, clinical inertia, multifaceted patient factors (including adherence to treatment, lifestyle and behaviors, and socioeconomic factors) and regional variances often associated with the wider determinants of health [32–35]. One final consideration in this context is the American Diabetes Association's recommendation to relax HbA1c treatment targets in older adults, allowing goals of 7.5% or even 8% to account for age-related limitations. These adjustments are made on a case-by-case basis. In this study, prevalence estimates were calculated using the universally accepted HbA1c threshold of 7% to ensure comparability with other published reports. However, under the extreme hypothetical assumption that all individuals aged 70 years and above were assessed using an 8% threshold, the prevalence of inadequately controlled diabetes in 2018 would decline by only 4% [24].

The findings found that male individuals with diabetes are at a significantly higher risk of having inadequately controlled T2DM compared to females. This risk was much higher after adjusting for the confounding effect of age, nationality, and comorbidities in a multivariate model. Conflicting evidence was reported in literature about the role of gender in diabetes control. A study in USA reported no sex difference in HbA1c control rates [36]. Another large sample national household survey in Bangladesh reported females having a higher prevalence of undiagnosed and uncontrolled diabetes [37].

The multivariate model showed a different role for age as a predictor of inadequately controlled T2DM. In the unadjusted bivariate model, extreme older age (70+years) significantly increased the risk of inadequately controlled T2DM. However, when adjusting for sex, nationality, and associated comorbidities, the highest risk was found in the youngest age group (18–39 years), with a 48% increase compared to the oldest age group. This suggests that the risk of inadequate control in older ages may be attributed to the higher prevalence of comorbidities. A recent study in Qatar supported this conclusion, attributing the higher prevalence of uncontrolled T2DM among the elderly to comorbidities and issues related to polypharmacy, which adversely impact adherence to treatment [6]. Interestingly a recent systematic review and mete-analysis reported no statistically significant association between age, gender among other tested predictors for uncontrolled diabetes. Whereas, increased physical activity was reported with decreased likelihood of poor glycemic control [14]. The higher risk of inadequate diabetes control among younger adults was also reported in a study from the USA, which was one of the first to highlight that younger adults had higher rates of uncontrolled hyperglycemia. Our study amplifies the call for further attention to the challenges of controlling diabetes in younger adults.[36].

The present study showed that diabetic patients with a South Asian ethnic background are at the highest risk of having inadequately controlled diabetes followed by Qatari nationals. These findings are substantiated with evidence indicating that immigrant South Asian populations have a high prevalence of type 2 diabetes mellitus are at a higher risk of developing inadequately controlled diabetes mellitus and associated health complications due to patient, provider and community level barriers [38–40]. Literature also suggests the influence of social determinants of health on the progression of type 2 diabetes mellitus [41].

The study reported a list of comorbidities associated with type 2 diabetes in a large population of diabetics over five years, with hypertension and dyslipidemia as the most frequently prevalent, affecting more than half of the analyzed population. Retinopathy ranked second in frequency at 17.3%. However, this study is not about the magnitude of comorbidities, but rather on their predictive capacity for the outcome. Being diagnosed with retinopathy increased the risk of inadequately controlled diabetes by 51%. This figure showed a vast increment to a 2.59 times increase in risk when the confounding effect of age, sex, and nationality was adjusted for in the multivariate model. This predictor was the most important among all other comorbidities, being second strongest after South-Asian nationalities.

## Study limitations

The association of comorbidities with inadequately controlled T2DM was interpreted as the risk factor leading to an outcome. This preferred interpretation can be challenged in a cross-sectional study which lacks directionality. Inadequately controlled T2DM for a certain duration can expedite the development of comorbidities and serve as a risk factor instead of an outcome. However, this does not limit the usefulness of comorbidities in identifying groups that require attention when designing T2DM management plan.

The time trend analysis of uncontrolled diabetes was not based on longitudinal paired data, but on multiple cross-sectional analyses. This type of analysis is prone for the confounding effect of personal attributes (like behavior, diet and exercise). This bias does not undermine the role of service delivery attributes in explaining the changes in diabetes control.

Selection bias may have affected the results of the study because of the 18.8% of censored data (individuals with a verified diagnosis of T2DM who did not have at least one HbA1C test result in a calendar year. This type of bias is non-directional and has a limited impact on the results, since it was less than 20% of missing data.

## Recommendations for future research and policy implications

The study findings can be utilized to design targeted interventions addressing the healthcare needs of high-risk patient groups. The perceptions, attitudes and health literacy of these groups can be further explored to determine the influence of broader determinants of health on the management and prevention of inadequately controlled T2DM. Additionally, designing health education campaigns should consider language barriers (inclusivity and cultural appropriateness) for various ethnic sub-groups with T2DM. Lastly, future research should address the confounding influence of pharmacotherapy and the categorization of T2DM patients into cohorts defined as newly controlled, newly uncontrolled, persistently uncontrolled, or persistently controlled.

## Author contributions

**Conceptualization:** Ahmed Sameer Alnuaimi, Muslim Abbas Syed, Mohamed Ahmed Syed.

**Data curation:** Ahmed Sameer Alnuaimi.

**Formal analysis:** Ahmed Sameer Alnuaimi.

**Funding acquisition:** Ahmed Sameer Alnuaimi.

**Investigation:** Ahmed Sameer Alnuaimi.

**Methodology:** Ahmed Sameer Alnuaimi.

**Project administration:** Ahmed Sameer Alnuaimi.

**Supervision:** Ahmed Sameer Alnuaimi.

**Writing – original draft:** Ahmed Sameer Alnuaimi, Muslim Abbas Syed, Mohamed Ahmed Syed.

**Writing – review & editing:** Ahmed Sameer Alnuaimi, Muslim Abbas Syed, Mohamed Ahmed Syed.

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
