## [Decision Letter · Decision Letter 0]

12 Nov 2025

Dear Dr. Alnuaimi,

Thank you for submitting your manuscript to PLOS ONE. After careful consideration, we feel that it has merit but does not fully meet PLOS ONE’s publication criteria as it currently stands. Therefore, we invite you to submit a revised version of the manuscript that addresses the points raised during the review process.

**ACADEMIC EDITOR:** - the introduction dosen't clearly include the justification for conducting this study in Qatar nor the rate of Diabetes in QatarMethod: please mention clearly under clear subheading; the study design, sample size calculation, number of participants=========================

We look forward to receiving your revised manuscript.

Kind regards,

Omnia S. El Seifi, M.D., Ph.D.

Academic Editor

PLOS ONE

“The Primary Health Care Corporation will cover the publication fees incurred by the journal.”

4. In the online submission form, you indicated that your data is available only on request from a third party. Please note that your Data Availability Statement is currently missing contact details for the third party, such as an email address or a link to where data requests can be made. Please update your statement with the missing information.

Reviewers' comments:

Reviewer's Responses to Questions

**Comments to the Author**

1. Is the manuscript technically sound, and do the data support the conclusions?

Reviewer #1: Yes

Reviewer #2: Yes

Reviewer #3: Partly

2. Has the statistical analysis been performed appropriately and rigorously?

Reviewer #1: Yes

Reviewer #2: Yes

Reviewer #3: I Don't Know

3. Have the authors made all data underlying the findings in their manuscript fully available?

Reviewer #1: Yes

Reviewer #2: Yes

Reviewer #3: No

4. Is the manuscript presented in an intelligible fashion and written in standard English?

Reviewer #1: Yes

Reviewer #2: No

Reviewer #3: Yes

Reviewer #1: REVIEWER’S COMMENTS: Prevalence and determinants of inadequately controlled diabetes in Qatar’s public primary care settings: A five-year cross-sectional review of health

records (2018-2022)

Dear Editor,

Thank you for inviting me to reviewer manuscript by Sameer et al on Prevalence and determinants of inadequately controlled diabetes in Qatar’s public primary care settings: A five-year cross-sectional review of health records (2018-2022).

Title and Abstract

Title: Clear and concise

Abstract: The abstract summarizes the study well, highlighting research problem and gaps clearly.

However, background should be summarized further and conclusion should be revise to include policy implication of findings, while deleting the recommendation as subheading.

Introduction: Well written and conceptualizes the research problem effectively. Nonetheless, there is need for revision to include literature review on prevalence of uncontrolled T2DM in middle east and other parts of the world.

Methodology: Methodology is generally sound and this will allow repeatability of the study and readers to quickly understand the study protocol. This helps establish validity and reliability of findings.

Results: The results are well presented and supported by appropriate statistical analysis. Moreover, there is need to revise first paragraph where the sentence starts with Of the eligible….. to maintain academic tone. HbA1C>=7% should be replace with standard abbreviation of ≥.

Tables and figures: Tables are well organized and figure caption described the information clearly however, correct the HbA1C>=7% to standard abbreviation of HbA1C ≥7%.

Discussion: The discussion is well structured and effectively compares the findings with previous research, highlighting limitations and recommendations.

Conclusion: The conclusion summarizes the study results effectively, including policy implication of the study.

References: The references are appropriate and relevant \, Moreover, ensure compliance with the journal guidelines.

Remarks: The study will contribute valuable scientific data considering a robust sample size from population-based study. Findings from this study may guide policy on relevant interventions to address the burden of uncontrolled DM and minimized complications.

Reviewer #2: Dear Author,

Thank you for the study on this topic of global relevance

Please find below my opinion and recommendation on your next review:

Ln1-3: Good topic with enough clarity but somehow too clumsy. Consider below suggestion or review to your own satisfaction:

"Prevalence and determinants of inadequately controlled diabetes in Qatar’s public primary care settings, a cross-sectional study."

Ln 19: Rather put " therefore is one of the ways"

Ln 20: Replace with "This"

Ln 22: Replace with " the risk of uncontrolled"

Ln 24: Replace with "review"

Ln 37: T2DM

Ln 40-42 : This is part of the conclusion of the study .

Proper recommendations may go to the body of the study. Thanks

Ln 55: put a comma. , followed by " as well as"

Ln 61/62 : These are not risk factors of T2DM but rather complications of untreated / poor glyceamic control in T2DM... Please review

Ln 62 :T2DM is a risk factor for coronary artery disease and cerebrovascular injury AND not the other way round... please review

Ln 79 -84 : Good descriptions of the study settings without ambiguity.. Thank you

Ln 85/86: Any particular reason why age is a criteria in this study, judging from the introduction where it was stated that the prevalence of T2DM is increasing in children and young adults and also it is a review of the records?

Please justify this.

Ln 121/122 : This is not in keeping with the earlier period of data extraction ( From 01/01/2018 to 31/12/2022)

Ln 144/145 : Was the data collection done before the ethical approval?

Ln 146 : It is always good to state if the extraction tools used have an implied consent that the patient information could be used for research purposes or if the patient registering with PHCC automatically consent to their records being used for research purposes.... if the later is applicable , please add to the study settings.

Ln 208/209 : This is interesting to see that there is a better glyceamic control during the Covid 19 pandemic.

Ln 228-231 : This study was actually conducted during the COVID19 Pandemic and most studies reported decline in the general control of chronic medical conditions, including poor glyceamic control in the T2DM due to multifactorial factors e.g inadequate drug dispensing resulting from "Lock downs" and cancelled clinic appointments due to movement restrictions .

This study seems not to be affected by Covid 19 pandemic despite the fact that it was conducted from 2018 till 2022.......What do you think about this?

Ln 279 : There are lot of limitations to this study . Look carefully into the methodology and the research question.

Also of note is the fact that comorbidities- outcome relationship in T2DM is routinely bidirectional cause/effect relationship in clinical medicine.

Reviewer #3: A generally well-written manuscript but methodologic questions as below. Could be considered for review by PLOS Global Public Health.

General:

--Beyond precedent in prior manuscripts, why was an A1c cut-off of 7% chosen? Higher A1c goals can be clinically considered in geriatric populations.

--Were individuals with anemia excluded since anemia can cause a falsely high A1c?

--Were pregnant individuals excluded since A1c levels generally decrease during pregnancy?

--any review of pharmacotherapy or diabetes incidence? if not, why not?

--some literature prefers T2D to T2DM; defer to authors preferences

--abstract and throughout (lines 175-176): While acknowledged in lines 280-285, inadequately controlled T2D is a clinical risk factor or predictor of retinopathy, not vice versa.

--throughout: ≥ rather than >=

--How does the study extend: https://www.gavinpublishers.com/article/view/factors-associated-with-glycemic-control-among-type-2-diabetic-patients-attending-primary-health-care-centers-in-qatar-a-cross-sectional-study ?

Lined:

--line 65: prefer adults with diabetes

--line 152: (Table 1)

--line 107: What is the rationale for "mean annual A1c" instead of categorizing as possible into newly controlled, newly uncontrolled, or persistently uncontrolled, or persistently controlled cohorts?

--lines 247-253: unclear with the distinctions here that seem to require dedicated statistical review of the models as they contribute to confusion throughout (e.g., lines 168-170 and Table 1 describe risk reduction but Table 2 shows increased ORs of inadequately controlled A1c; "Aging increases the risk of having inadequately controlled T2DM, but this was attributed to comorbidities. Adjusting for those comorbidities will show that younger ages are the ones at higher risk.")

**Do you want your identity to be public for this peer review?** For information about this choice, including consent withdrawal, please see our For information about this choice, including consent withdrawal, please see our Privacy Policy .

Reviewer #1: **Yes:** Dr Zainab AbdulkadirDr Zainab Abdulkadir

Reviewer #2: **Yes:** Adeloye Amoo Adeniji (MBBS; MMed; FCFP; FACRRM)Adeloye Amoo Adeniji (MBBS; MMed; FCFP; FACRRM)

Reviewer #3: No

---

## [Author Response · Author response to Decision Letter 1]

13 Jan 2026

reviewer's comment:

1. Good topic with enough clarity but somehow too clumsy. Consider below suggestion or review to your own satisfaction :

"Prevalence and determinants of inadequately controlled diabetes in Qatar’s public primary care settings, a cross-sectional study."

Author's reply: Excellent suggestion, which has been implemented in the manuscript.

2. Grammar comments made by the reviewer in the manuscript.

Author's reply: All the grammar corrections suggested by the reviewer were implemented in the reviewed manuscript.

3. Any particular reason why age is a criteria in this study, judging from the introduction where it was stated that the prevalence of T2DM is increasing in children and young adults and also it is a review of the records?

Authro's reply: Diabetes Mellitus in adolescents and youth is different from that in adults. I will refer to the results of the Treatment Options for Type 2 Diabetes in Adolescents and Youth (TODAY) study, which showed that type 2 diabetes is not only more aggressive in youth and adolescents than in adults, but complications from type 2 diabetes (T2D) are accelerated, and beta-cell decline happens faster. Therefore, the population of children warrants a separate study.

4. Referring to line 121 "The data extraction was completed on 122 10/01/2024 using custom made filters by the BHI". This is not in keeping with the earlier period of data extraction (From 01/01/2018 to 31/12/2022).

Author's reply: This manuscript was based on a data request project. Therefore, the date of data extraction would be much later than the actual data period.

5. Referring to line number 145 of the manuscript "(IRB reference number BUHOOTH-D-24-00055145 on 13/08/2023)". Was the data collection done before the ethical approval?

Author's reply: This is a data request research project. The actual data was generated as part of the routine PHCC health service transactions documented on CERNER (PHCC Electronic Health Record system). The deidentified data were curated by the Business Health Intelligence (BHI) department of PHCC. The IRB approval is provided to authorize the BHI to extract and deliver the anonymized data.

6. This study was actually conducted during the COVID19 Pandemic and most studies reported a decline in the general control of chronic medical conditions, including poor glyceamic control in the T2DM due to multifactorial factors e.g inadequate drug dispensing resulting from "Lock downs" and cancelled clinic appointments due to movement restrictions. This study seems not to be affected by Covid 19 pandemic despite the fact that it was conducted from 2018 till 2022.......What do you think about this?

Author's reply: It's an interesting observation. However, we prefer not to include it in the study, since it would provide a good example of "ecological fallacy" rather than a possible explanation for the improvement in glycemic control [SURVEY AND CORRELATIONAL RESEARCH DESIGNS. https://us.sagepub.com/sites/default/files/upm-binaries/57732_Chapter_8.pdf]. In addition, this incidental finding was not confined to the COVID period of 2020, but rather sustained afterwards. Please refer to this article [Nuaimi, A.S.A., Alam, M.T., Hassan, M. et al. Impact of COVID-19 restrictions on diabetes mellitus management in Qatari primary care settings. Discov Health Systems 3, 2 (2024). https://doi.org/10.1007/s44250-024-00065-x] that specifically studied the impact of COVID19 restrictions on DM control in a longitudinal paired design and showed that restrictions of COVID19 had a detrimental effect in our PHCC population.

7. It is always good to state if the extraction tools used have an implied consent that the patient information could be used for research purposes, or if the patient registering with PHCC automatically consent to their records being used for research purposes.... if the later is applicable , please add to the study settings.

Author's reply: Thank you for raising this point. Yes, Patients registering with PHCC automatically consent to their records being used for service auditing, quality improvement, and research purposes. This phrase is added to the manuscript.

8. Study limitations: There are a lot of limitations to this study. Look carefully into the methodology and the research question. Also of note is the fact that the comorbidities-outcome relationship in T2DM is a routinely bidirectional cause/effect relationship in clinical medicine.

Author's reply: Another two important limitations were added to the manuscript as listed below:

The time trend analysis of uncontrolled diabetes was not based on longitudinal paired data, but on multiple cross-sectional analyses. This type of analysis is prone for the confounding effect of personal attributes (like behavior, diet and exercise). This bias does not undermine the role of service delivery attributes in explaining the changes in diabetes control.

Selection bias may have affected the results of the study because of the 18.8% of censored data (individuals with a verified diagnosis of type II DM who did not have at least one HbA1C test result in a Calendar year. This type of bias is non-directional and has a limited impact on the results, since it was less than 20% of missing data.

9. Abstract: background should be summarized further and the conclusion should be revised to include policy implications of findings, while deleting the recommendation as a subheading.

Author's reply: Done

10. Introduction: There is a need for revision to include a literature review on the prevalence of uncontrolled T2DM in middle-east and other parts of the world.

Author's reply: The following paragraph was added to the manuscript: Despite the availability of adequately equipped health care systems in the six GCC (Gulf Cooperation Council) countries, inadequately controlled DM continues to pose an important health challenge. Literature reported figures between 54.9% in Qatar[19] to as high as 77% in KSA[20] for the prevalence rate of inadequately controlled diabetes.

11. Results: There is need to revise first paragraph where the sentence starts with Of the eligible….. to maintain academic tone.

Author's reply: We humbly request for an exemption to comply with this legible request. We will have to wait for up to six months of approvals to get this figure. As a work around this issue, the following statement was added to the manuscript (based on a previous data request) to serve the purpose" "This studied population represents 81.2% of all active (at least one clinical visit a year) PHCC registered population with a verified diagnosis of type II DM".

12. Results: HbA1C>=7% should be replace with standard abbreviation of ≥.

Author's reply: Done.

---

## [Decision Letter · Decision Letter 1]

11 Feb 2026

Dear Dr. Alnuaimi,

Thank you for submitting your manuscript to PLOS ONE. After careful consideration, we feel that it has merit but does not fully meet PLOS ONE’s publication criteria as it currently stands. Therefore, we invite you to submit a revised version of the manuscript that addresses the points raised during the review process.

We look forward to receiving your revised manuscript.

Kind regards,

Omnia S. El Seifi, M.D., Ph.D.

Academic Editor

PLOS One

Journal Requirements:

Reviewers' comments:

Reviewer's Responses to Questions

**Comments to the Author**

Reviewer #1: All comments have been addressed

Reviewer #2: All comments have been addressed

Reviewer #3: (No Response)

2. Is the manuscript technically sound, and do the data support the conclusions?

Reviewer #1: Yes

Reviewer #2: Yes

Reviewer #3: Partly

3. Has the statistical analysis been performed appropriately and rigorously?

Reviewer #1: Yes

Reviewer #2: Yes

Reviewer #3: I Don't Know

4. Have the authors made all data underlying the findings in their manuscript fully available?

Reviewer #1: Yes

Reviewer #2: Yes

Reviewer #3: No

5. Is the manuscript presented in an intelligible fashion and written in standard English?

Reviewer #1: Yes

Reviewer #2: Yes

Reviewer #3: Yes

Reviewer #1: Thank you for your efforts to improve the quality of the manuscript. I believe the significant changes have been made to the manuscript. Therefore, I recommend the manuscript should consider for publication.

Reviewer #2: Dear Author,

Thank you for prompt attention to my concerns in the first round of review

I think the study is now clear and well structured.

The data, strongly support the outcome of interest and the context, even though, most research at the same period of the study seems not to agree completely.

I think your study satisfy the minimum criteria for publication and it will provoke further studies on glyceamic control in type 2 DM. Good luck

Reviewer #3: The authors did not respond to many of the further below reviewer comments. Defer data availability appropriateness to editorial team.

--line 97: is there statistical rationale for this comorbidity score approach?

--line 107: acknowledge use of <7% as general A1c goal, but <7.5 or <8% can often be used as a goal in geriatric populations. 33% of this cohort was greater than or equal to 70 years old. can you complete any analyses in this demographic for goal A1c <7.5% or <8%? Fine for supplemental here..

--line 122: capitalization?

--line 296: T2DM

--line 297: no need to capitalize calendar

From prior (with updated line numbers):

General:

--Were individuals with anemia excluded since anemia can cause a falsely high A1c?

--Were pregnant individuals excluded since A1c levels generally decrease during pregnancy?

--any review of pharmacotherapy or diabetes incidence? if not, why not?

--abstract and throughout (lines 180-185): While acknowledged in lines 285-290, inadequately controlled T2D is a clinical risk factor or predictor of retinopathy (and chronic kidney disease and coronary disease), not vice versa.

--How does the study extend: https://www.gavinpublishers.com/article/view/factors-associated-with-glycemic-control-among-type-2-diabetic-patients-attending-primary-health-care-centers-in-qatar-a-cross-sectional-study ?

Lined:

--line 63: prefer adults with diabetes

--line 155: (Table 1)

--line 109: What is the rationale for "mean annual A1c" instead of categorizing as possible into newly controlled, newly uncontrolled, or persistently uncontrolled, or persistently controlled cohorts?

--lines 252-258: unclear with the distinctions here that seem to require dedicated statistical review of the models as they contribute to confusion throughout (e.g., lines 173-175 and Table 1 describe risk reduction but Table 2 shows increased ORs of inadequately controlled A1c; "Aging increases the risk of having inadequately controlled T2DM, but this was attributed to comorbidities. Adjusting for those comorbidities will show that younger ages are the ones at higher risk.")

**Do you want your identity to be public for this peer review?** For information about this choice, including consent withdrawal, please see our For information about this choice, including consent withdrawal, please see our Privacy Policy .

Reviewer #1: **Yes:** Dr Zainab Abdulkadir (MBBS,MSc PH, FMCFM, MD)Dr Zainab Abdulkadir (MBBS,MSc PH, FMCFM, MD)

Reviewer #2: **Yes:** Adeloye Amoo Adeniji (MBBS; MMed; FCFP; FACRRM)Adeloye Amoo Adeniji (MBBS; MMed; FCFP; FACRRM)

Reviewer #3: No

---

## [Author Response · Author response to Decision Letter 2]

18 Feb 2026

Reviewer #3: The authors did not respond to many of the further below reviewer comments. Defer data availability appropriateness to editorial team.

--line 97: is there statistical rationale for this comorbidity score approach?

Author’s response: The comorbidity score is a non-biased statistical approach for studying the burden of comorbid conditions in a diabetic individual. The selected comorbidities are known to occupy the highest rank in a long list of comorbid conditions associated with DM. In addition, the selected conditions can be easily and accurately captured as diagnostic codes by the managing physician in the electronic health record system of the PHCC.

--line 107: acknowledge use of <7% as general A1c goal, but <7.5 or <8% can often be used as a goal in geriatric populations. 33% of this cohort was greater than or equal to 70 years old. can you complete any analyses in this demographic for goal A1c <7.5% or <8%? Fine for supplemental here..

Author’s response: This is an interesting comment. Relaxing the HbA1c cut-off value for proper DM control to 7.5% or 8% is decided on a case-by-case basis for those older than 70 years depending on their overall health status. This is not an absolute requirement, but realistic accommodation for the compromised capacity of older age group. Including such an adjustment in calculating the population level prevalence of inadequately controlled diabetes would introduce bias and limit the comparison with other studies. In addition, since the focus of this study is to portray the primary care system performance in managing DM it would be more conservative to use the global HbA1c criteria of <7%.

To satisfy our scientific curiosity the table below will demonstrate the trivial impact of adjusting the cut-off value for older ages. For example, increasing the cut-off value of HbA1c for defining inadequately controlled DM in older ages (70+ years) to 7.5% would reduce the overall prevalence rate by 2% in the year 2018 compared to the global estimate. In addition, further increasing the cut-off value to 8% would reduce the prevalence rate by 4% in the year 2018.

Prevalence rate of Inadequately controlled DM by year 2018 2019 2020 2021 2022

Global criteria (HbA1C>=7%) 58.8 55.3 53.2 53.5 53.9

Adjusted for older age (70+ years) exception (HbA1C>=7%, except for ages 70+ [>=7.5%]) 56.5 53.1 51.3 51.9 52.4

Adjusted for older age (70+ years) exception (HbA1C>=7%, except for ages 70+ [>=8%]) 54.5 51.4 49.9 50.6 51.2

--line 122: capitalization?

Author’s response: Done.

--line 296: T2DM

Author’s response: Done.

--line 297: no need to capitalize calendar

Author’s response: Done.

From prior (with updated line numbers):

General:

--Were individuals with anemia excluded since anemia can cause a falsely high A1c?

Author’s response: No exclusion of anemia was practiced. We will report this as a study limitation.

--Were pregnant individuals excluded since A1c levels generally decrease during pregnancy?

Author’s response: Pregnant females were excluded. We will update this in the study population.

--any review of pharmacotherapy or diabetes incidence? if not, why not?

Author’s response: I’m sorry we did not understand the comment correctly. Why do we need a review of pharmacotherapy or diabetes incidence to describe the epidemiology of inadequately controlled DM. The focus of the manuscript is the health system and primary care in specific and not the type of treatment and its performance. Nevertheless, It’s a good suggestion for a new research idea about DM.

--abstract and throughout (lines 180-185): While acknowledged in lines 285-290, inadequately controlled T2D is a clinical risk factor or predictor of retinopathy (and chronic kidney disease and coronary disease), not vice versa.

Author’s response: The term “RISK” is a statistical term referring to the strength of association between an explanatory and an outcome variables. It’s written as risk but refers to probability. This study is cross-sectional which fails to show a directional relationship between the variables. That is why statistical risk is a measure of effect size and is different from clinical risk in which the cause precedes the effect. However, the term risk was changed to probability to reduce the confusion.

--How does the study extend: https://www.gavinpublishers.com/article/view/factors-associated-with-glycemic-control-among-type-2-diabetic-patients-attending-primary-health-care-centers-in-qatar-a-cross-sectional-study ?

Author’s response: The study referred to can be considered as a small pilot project with convenient sample that highlighted the importance of uncovering the reasons behind having a substantial size of uncontrolled DM. The current study included a huge study population, which can describe the inadequately controlled DM at the wider primary health care system.

Lined:

--line 63: prefer adults with diabetes

Author’s response: Done.

--line 155: (Table 1)

Author’s response: Done.

--line 109: What is the rationale for "mean annual A1c" instead of categorizing as possible into newly controlled, newly uncontrolled, or persistently uncontrolled, or persistently controlled cohorts?

Author’s response: It’s a known statistical assumption that mean is the most unbiased representation of a sample of quantities that are known or assumed to be normally distributed. Since HbA1c is a cumulative representation of blood glucose throughout a period of at least 3 months, it is unlikely for the two or probably measurements taken during a year to show extreme fluctuations that may bias the calculation of the mean. The mean HbA1c qualifies as a good index representing diabetes control over a year. The suggested terminologies of newly controlled, newly uncontrolled, or persistently uncontrolled, or persistently controlled cohorts may look plausible, but are difficult to defend and justify when a specific cut-off value is addressed.

--lines 252-258: unclear with the distinctions here that seem to require dedicated statistical review of the models as they contribute to confusion throughout (e.g., lines 173-175 and Table 1 describe risk reduction but Table 2 shows increased ORs of inadequately controlled A1c; "Aging increases the risk of having inadequately controlled T2DM, but this was attributed to comorbidities. Adjusting for those comorbidities will show that younger ages are the ones at higher risk.")

Author’s response: We have consulted a biostatistician to review the results of multivariate models before releasing the manuscript for a scientific publication. Can you please clarify what is needed from our end?

---

## [Decision Letter · Decision Letter 2]

4 Mar 2026

Dear Dr. Alnuaimi,

Thank you for submitting your manuscript to PLOS ONE. After careful consideration, we feel that it has merit but does not fully meet PLOS ONE’s publication criteria as it currently stands. Therefore, we invite you to submit a revised version of the manuscript that addresses the points raised during the review process.

We look forward to receiving your revised manuscript.

Kind regards,

Omnia S. El Seifi, M.D., Ph.D.

Academic Editor

PLOS One

**Journal Requirements:**

Reviewers' comments:

Reviewer's Responses to Questions

**Comments to the Author**

Reviewer #3: (No Response)

2. Is the manuscript technically sound, and do the data support the conclusions?

Reviewer #3: Yes

3. Has the statistical analysis been performed appropriately and rigorously?

Reviewer #3: I Don't Know

4. Have the authors made all data underlying the findings in their manuscript fully available?

Reviewer #3: No

5. Is the manuscript presented in an intelligible fashion and written in standard English?

Reviewer #3: Yes

**Reviewer #3:**  Appreciate the authors' thoughtful responses that clarify. Some minor comments as below. Appreciate the authors' thoughtful responses that clarify. Some minor comments as below.

--Would suggest mentioning negligible effects in analyses of higher geriatric A1c goals in the discussion

--Pharmacotherapy and T2D incidence (and newly controlled, newly uncontrolled, or persistently uncontrolled, or persistently controlled cohorts) were suggested to clarify as able if "inadequately controlled T2D" was related to newly diagnosed uncontrolled diabetes that subsequently became controlled rather than chronically uncontrolled diabetes.. Fine not to include per authors' response; fine to include in limitations or future directions if authors think helpful..

--For participants with three or more A1c values in a year, were A1c values normally distributed in each of the five years? This seems that it would most best justify the role of the mean annual A1c ("three or more values" to allow a limited within-participant normality test)

-line 66: six

-line 96 and Table 1: "comorbidity score" description could be deleted and Table 1 just state "number of comorbidities" to more directly describe approach but fine to defer to authors' preferences

-line 147: why highlighted?

-line 156, 298: T2DM

Defer 1) data availability appropriateness and 2)statistical review of the below to the journal editorial team.

--lines 254-260: unclear with the distinctions here that seem to require dedicated statistical review of the models as they contribute to confusion throughout (e.g., lines 172-174 and Table 1 describe risk reduction but Table 2 shows increased ORs of inadequately controlled A1c; "Aging increases the risk of having inadequately controlled T2DM, but this was attributed to comorbidities. Adjusting for those comorbidities will show that younger ages are the ones at higher risk.")

**Do you want your identity to be public for this peer review?** For information about this choice, including consent withdrawal, please see our For information about this choice, including consent withdrawal, please see our Privacy Policy .

Reviewer #3: No

---

## [Author Response · Author response to Decision Letter 3]

5 Mar 2026

Reviewer #3: 6. Review Comments to the Author

Reviewer #3: Appreciate the authors' thoughtful responses that clarify. Some minor comments as below.

-Would suggest mentioning negligible effects in analyses of higher geriatric A1c goals in the discussion

Author’s response: The below paragraph was added to the first part of discussion to address the reviewer’s comment:

One final consideration in this context is the American Diabetes Association’s recommendation to relax HbA1c treatment targets in older adults, allowing goals of 7.5% or even 8% to account for age-related limitations. These adjustments are made on a case-by-case basis. In this study, prevalence estimates were calculated using the universally accepted HbA1c threshold of 7% to ensure comparability with other published reports. However, under the extreme hypothetical assumption that all individuals aged 70 years and above were assessed using an 8% threshold, the prevalence of inadequately controlled diabetes in 2018 would decline by only 4%[24].

-Pharmacotherapy and T2D incidence (and newly controlled, newly uncontrolled, or persistently uncontrolled, or persistently controlled cohorts) were suggested to clarify as able if "inadequately controlled T2D" was related to newly diagnosed uncontrolled diabetes that subsequently became controlled rather than chronically uncontrolled diabetes. Fine not to include per authors' response; fine to include in limitations or future directions if authors think helpful.

Authors response: This new paragraph was added to the “Recommendations section”:

Lastly, future research should address the confounding influence of pharmacotherapy and the categorization of T2DM patients into cohorts defined as newly controlled, newly uncontrolled, persistently uncontrolled, or persistently controlled.

-For participants with three or more A1c values in a year, were A1c values normally distributed in each of the five years? This seems that it would most best justify the role of the mean annual A1c ("three or more values" to allow a limited within-participant normality test)

Authors response: Thank you for posting this comment. Let me refer to a known statistical theory called the “central limit theorem”. It states that the mean will always be as close as possible to a normal distribution for very large samples, which is the case in this study.

-line 66: six

Authors response: Done

-line 96 and Table 1: "comorbidity score" description could be deleted and Table 1 just state "number of comorbidities" to more directly describe approach but fine to defer to authors' preferences

Authors response: The new version of the statement is as follows:

Summing positive comorbidities per individual created a new variable (number of comorbidities) to reflect the health burden.

The “comorbidity score” (in table 1) was changed to “number of comorbidities”.

-line 147: why highlighted?

Authors response: Honest oversight.

-line 156, 298: T2DM

Authors response: Done

---

## [Decision Letter · Decision Letter 3]

17 Mar 2026

Prevalence and determinants of inadequately controlled diabetes in Qatar’s public primary care settings: A cross-sectional study

PONE-D-25-42745R3

Dear Dr. Alnuaimi,

We’re pleased to inform you that your manuscript has been judged scientifically suitable for publication and will be formally accepted for publication once it meets all outstanding technical requirements.

Kind regards,

Omnia S. El Seifi, M.D., Ph.D.

Academic Editor

PLOS One

Additional Editor Comments (optional):

Reviewers' comments:

Reviewer's Responses to Questions

**Comments to the Author**

Reviewer #3: All comments have been addressed

2. Is the manuscript technically sound, and do the data support the conclusions?

Reviewer #3: Yes

3. Has the statistical analysis been performed appropriately and rigorously?

Reviewer #3: I Don't Know

4. Have the authors made all data underlying the findings in their manuscript fully available?

Reviewer #3: No

5. Is the manuscript presented in an intelligible fashion and written in standard English?

Reviewer #3: Yes

Reviewer #3: All comments have been thoughtfully addressed.

As prior, defer 1) data availability appropriateness and 2) statistical review of the below to the journal editorial team.

--lines 254-260: unclear with the distinctions here that seem to require dedicated statistical review of the models as they contribute to confusion throughout (e.g., lines 172-174 and Table 1 describe risk reduction but Table 2 shows increased ORs of inadequately controlled A1c; "Aging increases the risk of having inadequately controlled T2DM, but this was attributed to comorbidities. Adjusting for those comorbidities will show that younger ages are the ones at higher risk.")

**Do you want your identity to be public for this peer review?** For information about this choice, including consent withdrawal, please see our For information about this choice, including consent withdrawal, please see our Privacy Policy .

Reviewer #3: No

---

## [Editor Report · Acceptance letter]

PONE-D-25-42745R3

PLOS One

Dear Dr. Alnuaimi,

I'm pleased to inform you that your manuscript has been deemed suitable for publication in PLOS One. Congratulations! Your manuscript is now being handed over to our production team.

Kind regards,

on behalf of

Professor Omnia S. El Seifi

Academic Editor

PLOS One